# Serological Surveillance of Influenza D Virus in Ruminants and Swine in West and East Africa, 2017–2020

**DOI:** 10.3390/v13091749

**Published:** 2021-09-02

**Authors:** Idrissa Nonmon Sanogo, Casimir Kouakou, Komla Batawui, Fidélia Djegui, Denis K. Byarugaba, Rachidatou Adjin, Komlan Adjabli, Fred Wabwire-Mangen, Bernard Erima, Gladys Atim, Qouilazoni A. Ukuli, Titus Tugume, Koffi Dogno, Wolali Go-Maro, Emmanuel Couacy-Hymann, Ghazi Kayali, Pamela McKenzie, Richard J. Webby, Mariette F. Ducatez

**Affiliations:** 1Interactions Hôtes-Agents Pathogènes (IHAP), UMR 1225, Université de Toulouse, ENVT, INRAE, 31076 Toulouse, France; idrissa.sanogo@envt.fr; 2Faculté d’Agronomie et de Médecine Animale (FAMA), Université de Ségou, Ségou BP 24, Mali; 3Laboratoire National d’Appui au Développement Agricole (LANADA), Bingerville BP 206, Côte d’Ivoire; casymyr2006@yahoo.fr (C.K.); chymann@gmail.com (E.C.-H.); 4Laboratoire Central Vétérinaire de Lomé, Lomé BP 4041, Togo; dbatawui@yahoo.fr (K.B.); fo_mathias@yahoo.fr (K.A.); koffipolo001@yahoo.fr (K.D.); emilygomaro@yahoo.fr (W.G.-M.); 5Laboratoire de Diagnostic vétérinaire et de Sérosurveillance (LADISERO), Parakou BP 23, Benin; djegui_fidelia@yahoo.fr (F.D.); adjinrachidath@yahoo.fr (R.A.); 6College of Veterinary Medicine Animal Resources and Biosecurity, Makerere University, Kampala P.O. Box 7062, Uganda; denis.byarugaba@mak.ac.ug (D.K.B.); fwabwire@musph.ac.ug (F.W.-M.); 7Makerere University Walter Reed Project, Kampala P.O. Box 16524, Uganda; berima@muwrp.org (B.E.); gatim@muwrp.org (G.A.); qukuli@muwrp.org (Q.A.U.); ttugume@muwrp.org (T.T.); 8Human Link, Dubai Multi Commodity Center (DMCC), Dubai 48800, United Arab Emirates; ghazi@human-link.org; 9Health Sciences Center, University of Texas, Houston, TX 77030, USA; 10Department of Infectious Diseases, St. Jude Children’s Research Hospital, Memphis, TN 38105, USA; Pamela.McKenzie@stjude.org (P.M.); richard.webby@stjude.org (R.J.W.)

**Keywords:** influenza D virus, Africa, cattle, small ruminants, swine, epidemiology, serology

## Abstract

Influenza D virus (IDV) was first isolated in 2011 in Oklahoma, USA from pigs presenting with influenza-like symptoms. IDV is known to mainly circulate in ruminants, especially cattle. In Africa, there is limited information on the epidemiology of IDV, although the virus has likely circulated in the region since 2012. In the present study, we investigated the seropositivity of IDV among domestic ruminants and swine in West and East Africa from 2017 to 2020. Serum samples were analyzed using the hemagglutination inhibition (HI) assay. Our study demonstrated that IDV is still circulating in Africa, with variations in seropositivity among countries and species. The highest seropositivity was detected in cattle (3.9 to 20.9%). Our data highlights a need for extensive surveillance of IDV in Africa in order to better understand the epidemiology of the virus in the region.

## 1. Introduction

Influenza viruses are members of the *Orthomyxoviridae* family and are divided into four types (A, B, C, D). Animals are mainly infected by influenza type A and D viruses [1,2]. Influenza D Virus (IDV) was first isolated in 2011 in Oklahoma, USA from pigs presenting with flu-like symptoms [3]. After its first isolation, IDV has been detected in other countries in America [4], Europe [5,6,7,8] and Asia [9,10]. Serological studies showed the circulation of IDV in many countries in Africa [11,12,13], suggesting a worldwide geographic distribution of the virus. 

Evidence for IDV infection has been found in a large number of animal species, such as cattle, sheep, goats [14,15], pigs and wild boars [16], dromedary camels [12] and horses [17]. Cattle are the main host of IDV, contributing to the spread of the virus to other animal species [5,18,19]. IDV is considered to play a role in the bovine respiratory disease complex [4], which is responsible for huge economic losses for the livestock sector globally. In addition to animals, IDV-specific antibodies have been found in humans, with very high seroprevalence among people exposed to cattle, thus suggesting the zoonotic potential of the virus [20,21]. 

In Africa, there is limited information on the epidemiology of IDV, although evidence suggests that the virus may have been circulating in the region since at least 2012 [11]. Studies in Kenya and Ethiopia [11,12] showed substantially higher seroprevalence in camelids (99%), suggesting the importance of the species in the epidemiology of IDV in Africa. In Morocco (North Africa), 35% of cattle sera collected were positive for anti-IDV antibodies [11]. In West Africa, lower seropositivity rates for IDV were reported in cattle (1.9–10.4%) and small ruminants (1.4–2.2%) [11]. To date, no antibodies were detected in pigs in Africa, and attempts to isolate the virus have been unsuccessful [11,13].

The capacity of IDV to infect humans and numerous other animal species, and the ability of influenza viruses to regularly mutate, make this virus a continuing threat to animal and public health. Therefore, it is necessary to carry out further research on this virus. In this study, we aimed to investigate the seropositivity of IDV among domestic ruminants and swine in West and East Africa from 2017 to 2020 and to compare these results with previously available data.

## 2. Materials and Methods

### 2.1. Serum Samples Collection

From 2017 to 2020, we conducted a prospective serosurveillance for IDV in four countries in Sub-Saharan West and East Africa (Figure 1). A total of 3381 sera were collected from apparently healthy adult domestic ruminants and swine either from farms, cattle markets or slaughterhouses. Immediately after collection, the samples were stored in a cool box at +4 °C and transferred to the laboratory within 24 h. Animal health status and sex, as well as the date and place of sampling, were recorded. Sera samples collected in Togo during a previous study on IDV were also included in this study [13].

### 2.2. Serological Analysis

For the detection of IDV antibodies in serum, hemagglutination inhibition (HI) tests were performed using D/bovine/France/5920/2014 as the antigen and 1% chicken red blood cells for hemadsorption according to the procedures described by Salem et al. [11]. The HI titer was expressed as the reciprocal of the highest serum dilution showing complete inhibition of hemagglutination. A serum sample was determined as seropositive when the HI titer was ≥1: 10 [11,22].

### 2.3. Data Analysis

Data were recorded into a standard Excel spreadsheet in Microsoft Excel 2016 (Microsoft corporation, Seattle, WA, USA). The anti-IDV seropositivity in each country and animal species was calculated as the number of sera with a titer greater than or equal to 1:10 out of the total number of sera tested. Statistical analyses were carried out with RStudio Desktop 1.4 software (RStudio Inc., Boston, MA, USA). The Chi-square test of independence was performed to test the relationship between the prevalence of anti-IDV antibodies and different explanatory variables (species, sex and place of sampling) at a level of significance of 5% (*p* ≤ 0.05). We then examined the effects of species, countries and sex, individually, on the odds of seropositivity to IDV by using the epi.2by2 function of the epiR package in RStudio Desktop 1.4 (RStudio, Inc., Boston, MA, USA).

## 3. Results

### 3.1. Seropositivity of IDV in Cattle, Sheep, Goats and Swine

The results of HI analyses revealed that anti-IDV antibodies were present in a percentage of serum samples from all four animal species (Table 1). Out of the 3381 samples collected, 232 (6.9%) were positive for IDV antibodies. The highest seropositivity rate for IDV (20.9 %) was observed in cattle in Uganda. Seropositive swine were also detected only in Uganda. Sheep and goats had much lower seropositivity (2 to 4.4%) than did cattle. HI titers observed in positive samples ranged from 10 to 640, with the highest titers seen in cattle and swine sera. When selecting 20 instead of 10 as a threshold titer for positive sera, very similar results were obtained.

### 3.2. Factors Associated with IDV Seropositivity 

We next determined what factors were associated with IDV seropositivity. The variables that had a statistically significant association with IDV seropositivity are presented in Table 2. Cattle were more likely to be seropositive than goats or sheep (OR = 0.48 for goats and OR = 0.30 for sheeps *p* < 0.001); however, swine had 1.5 times higher odds (OR= 1.56, *p* = 0.03) of being seropositive to IDV in comparison to cattle. Additionally, cattle in Uganda had a higher risk of being infected by IDV than in other countries (OR 5.33, *p* < 0.001).

There was no significant difference between seropositivity in cattle in Benin, Côte d’Ivoire and Togo. Moreover, there was no statistically significant difference in seropositivity by sex.

## 4. Discussion

In this study, we determined the prevalence of anti-IDV antibodies among domestic ruminants and swine in West and East Africa. Our results confirm that IDV is still circulating in the region, as previously shown by other studies in Africa [11,13]. However, in Benin and Côte d’Ivoire, our data showed higher IDV seropositivity in cattle than seen in prior studies assessing sera collected between 1991–2015 (1.9% and 0%, respectively). Conversely, IDV seropositivity in cattle in Togo in our study was lower than that seen in sera collected from 1991–2015 (10.4%). We did find a higher IDV seropositivity rate in cattle in Togo (6.3 %) compared with data previously reported between 2017–2019 (4.5%), suggesting that the virus continues to circulate widely. 

IDV seropositivity in cattle differed significantly between countries in West and East Africa. These differences between countries might be related to the variability in husbandry practices, as previously suggested [14].We found that Uganda had the highest IDV seropositivity rate in cattle. This finding may be the result of the importance of the cattle population in Uganda (16.3 million heads in 2019) compared to the other three countries in West Africa (Benin: 2.5 million; Côte d’Ivoire: 1.7 million; Togo: 0.5 million heads) [23]. The higher seropositivity in Uganda might also be partly explained by the high number of camels in East Africa (60% of the world population in the horn of Africa), as camels have been postulated to contribute to the persistence of IDV in Africa [12,24]. Nevertheless, further serological and virological studies are required to better understand the role played by camels in the epidemiology of IDV.

Seropositivity rates of IDV in domestic ruminants and swine and HI titers in Africa are much lower than those observed in other continents [8,14,19]. Antigenically different IDV strains were isolated in many regions of the world [5]. In this study, the virus we used for the HI test was D/bovine/France/5920/2014, isolated in France, and classified as a D/swine/Oklahoma-like virus. It cannot be ruled out that antigenically different IDV strains may be circulating in Africa, lowering the sensitivity of our assay.

Our data indicated a higher IDV seropositivity rate in cattle compared to the other animal species. These results are in line with data reported in European countries [8]. Indeed, cattle are considered as the main host for IDV, contributing to its spread. Furthermore, high levels of IDV antibodies were reported in newborn calves [19]; our study focused only on adult animals as they are the ones found in livestock markets and slaughterhouses.

Our study provides the first serological evidence of IDV in swine in Africa. Previous studies have reported IDV in swine in France [16] and Italy [6]. However, IDV is circulating at lower levels in swine compared with cattle, but at higher levels than in sheep [15]. Our study found higher odds for swine to be seropositive to IDV when compared with cattle. However, the number of swine sera was limited and only collected in Togo and Uganda, underpowering our study to some degree.

The results presented in this study, while commendable, should be treated cautiously because they are based on convenience samples. In addition, we did not carry out any confirmatory serological tests, such as ELISA or microneutralization assays. Thus, large-scale and well-designed studies based on random samples are needed to expand on our results.

In summary, our findings confirm that IDV is circulating among domestic animals in West and East Africa, albeit at lower rates than observed in other regions. We identified evidence of variation in IDV seropositivity among different countries and species. Nevertheless, as our study was based on non-random samples, further serological and virological studies are needed to investigate the epidemiology of IDV in Africa and to examine its potential threat to public health.

## Figures and Tables

**Figure 1 viruses-13-01749-f001:**
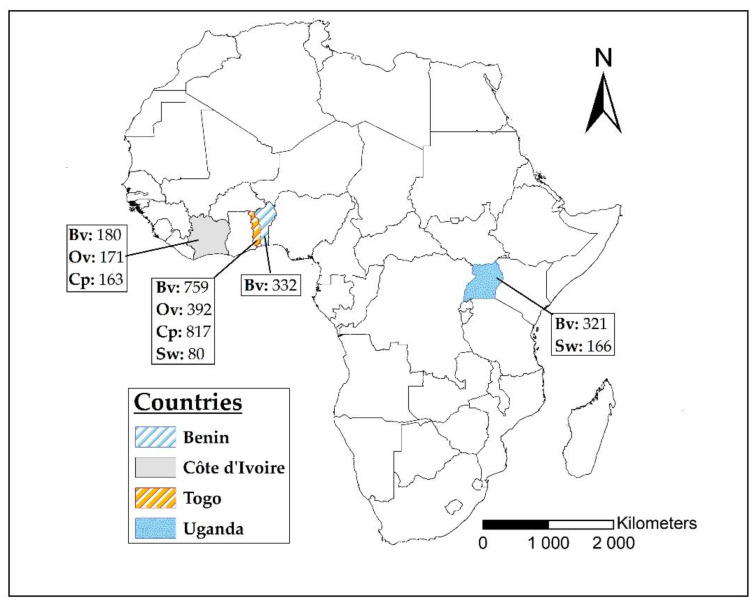
Map of countries where sera were collected. The number of samples collected from each species is indicated. Bv: cattle; Ov: sheep; Cp: goat; Sw: swine. In Uganda, all sera were collected in slaughterhouses. The map was designed using ArcMap 10.4 (ESRI, Redlands, CA, USA).

**Table 1 viruses-13-01749-t001:** Seropositivity rates of influenza D virus (IDV) in cattle, sheep, goat and swine in four countries of West and East Africa.

Country	Sampling Period	Animal Species	Number of Tested Samples	Number of Positive Samples	Positivity Rate (%)	HI Titer Range
Benin	2017–2019	Cattle	332	13	3.9	20–80
Côte d’Ivoire	2019	Cattle	180	13	7.2	10–80
		Sheep	171	7	4.1	10–40
		Goat	163	6	3.7	10–40
Togo	2017–2020	Cattle	759	48	6.3	10–320
		Sheep	392	8	2	10–80
		Goat	817	36	4.4	20–160
		Swine	80	0	–	–
Uganda	2017–2019	Cattle	321	67	20.9	10–160
		Swine	166	34	20.5	10–640

**Table 2 viruses-13-01749-t002:** Factors associated with anti-IDV seropositivity rates among cattle, sheep, goat and swine in four countries of West and East Africa.

Variable	Categories (N)	n (%)	OR	95% (CI)	*p*
Species	Cattle (1592)	141 (8.9)	RF		
	Goat (980)	42 (4.3)	0.48	0.34–0.69	<0.001
	Sheep (563)	15 (2.7)	0.30	0.18–0.52	<0.001
	Swine (246)	34 (13.6)	1.56	1.05–2.32	0.03
Countries *	Benin (332)	13 (3.9)	RF		
	Côte d’Ivoire (180)	13 (7.2)	–	–	0.1
	Togo (759)	48 (6.3)	–	–	0.1
	Uganda (321)	67 (20.9)	5.33	2.89–9.84	<0.001
Sex	Male (1456)	90 (6.2)	RF		
	Female (1925)	142 (7.4)	–	–	0.2

N: number of samples tested; n: number of positive samples; %: proportion of positive samples; OR: odds ratios; CI: confidence interval; RF: reference factor; *p* values ≤ 0.05 are considered statistically significant; * For these countries, only cattle sera were considered.

## Data Availability

The data presented in this study are available upon request to the corresponding author.

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
