# Peer review of "Serological Surveillance of Influenza D Virus in Ruminants and Swine in West and East Africa, 2017–2020"

_viruses, 2021, doi:10.3390/v13091749_

Round 1

Reviewer 1 Report

In this brief report, Sanogo et al. performed an IDV serological surveillance for cattle, goat, sheep, and swine in four African countries,  Bernin, Cote dIvoire, Togo, and Uganda. This study involved a total of 3381 samples from 2017 to 2020. The authors identified IDV seropositivity varied among all four countries and among animal species. This study is a continued serosurveillance study by the same team, and their results showed a dynamics of IDV seropositivity rates in the same country across years, even for the same animal species.  Importantly, this study reported the seropositive swine samples in Uganda. Overall, this study is significant and facilitated our understanding of the IDV natural history, particularly in the animal population in Africa, which is understudied.

I hope the following two minor comments can help improve this manuscript. 1) It will be beneficial to add more background for sampling herds, for example, a table to summarize the sampling details such as herd samples, sample size, types of herd, and so on. This can help us understand the total size of animal herds versus the number of samples collected as well as other epidemiological attributes. 2) When perform statistical analyses as well as OR of IDV infection, it will be more informative to show seropositive rate per group (or herds) samples (instead of grouping all of them from the same country). Otherwise, OR for infection will not be informative.

Author Response

1) It will be beneficial to add more background for sampling herds, for example, a table to summarize the sampling details such as herd samples, sample size, types of herd, and so on. This can help us understand the total size of animal herds versus the number of samples collected as well as other epidemiological attributes.

Answer: We agree with the Reviewer that adding this type of epi data would make a lot of sense and improve the understanding of the sampling context. Unfortunately, samples were mainly collected from cattle markets and slaughterhouses where pieces of information on herds are missing irrespective of the country. We however have the number of samples collected per site (without knowing the total number of cattle/small ruminant/swine heads per site): 30 - 60 per month in Uganda, 20 – 120 per month in Togo, 20 - 40 per month in Benin and 60 – 105 in Côte d’Ivoire. As we miss the numbers of herds, animals per herd, and total number of animals per market or slaughterhouse, we feel it would not be very informative to add these samples number in the text.

We have indicated that we went for convenient sampling in the Discussion section to highlight the absence of randomization, and the poor representatively of the whole countries animal populations here.

2) When perform statistical analyses as well as OR of IDV infection, it will be more informative to show seropositive rate per group (or herds) samples (instead of grouping all of them from the same country). Otherwise, OR for infection will not be informative.

Answer: We agree with the Reviewer that it would have been more informative to calculate the seropositive rate and the OR per herds instead of grouping them. As explained above, our samples were collected from cattle markets and slaughterhouses and we were not able to find data at herds level. Thus, it was not possible for us to determine the seropositivity rate by herds.

However, for each country we did calculate the seropositivity rate by species and we think that this piece of information gives us a good overview of the difference of anti-IDV antibodies circulation among countries and species. With the data available, we were able to examine the relationship between countries, species and sex and the seropositivity rate of IDV by calculating the OR. However, for the OR between countries, we have considered only cattle sera as these were available for the 4 countries of study and we explained this in the manuscript (line 128).

Reviewer 2 Report

The paper by Nonmon Sanogo and co-authors reported the serological evidence of circulation of IDV in Africa in several species, ruminants and swine, the last few years. The manuscript is well written and scientifically sound for publication. It is appreciable to point out in the discussion the absence of randomized sampling for the design of the study. A few minor suggestions should be addressed.

In the Materials and Methods section, the authors may explain the pre-treatment of the sera line 82. In the reference paper (Salem et al), the sera were hemadsorbed on red blood cells from horse. This is probably not the case here.

In the Results section, there is a difference between the OR line 119 (6.46) and that of the table 2 (6.47). Line 121 and in the table 2, the name Ivoire for the country is missing a capital letter. The authors mentionned they collected the age of the tested animals but it was not taken into account as a factor. Would it be interesting to study despite the fact that the population tested was adult ? 

In the Discussion section, line 136, the authors may add "previously" to better understand the comparison of the studies from the same period: ...in Togo (6.3%) compared to data previously reported between 2017-2019 (4.5%)...

Author Response

1) In the Materials and Methods section, the authors may explain the pre-treatment of the sera line 82. In the reference paper (Salem et al), the sera were hemadsorbed on red blood cells from horse. This is probably not the case here.

Answer: We agree with the Reviewer and apologize for the oversight. We have used chicken red blood cells for hemadsorption in the present study. The text has been modified to read as follows (lines 84-85):
………..to the procedures described by Salem et al. [11] but hemadsorption was performed with chicken red blood cells.

2) In the Results section, there is a difference between the OR line 119 (6.46) and that of the table 2 (6.47).

Answer: We thank the Reviewer for pointing this out. We have repeated the statistical analysis and we have found some minor errors, which did not significantly change our results. Thus, we have updated the table 2 and we have made changes to the results (line 116 - 120) and to the discussion sections (lines 165 - 166). It reads as follows:

Results section:

“Cattle were more likely to be seropositive than goats or sheep (OR=0.48 for goat and OR=0.30 for sheep P<0.001); however, swine had 1.5 higher odds (OR= 1.56, p = 0.03) to be seropositive to IDV in comparison to cattle. Additionally, cattle in Uganda had higher risk to be infected by IDV than in other countries (OR 5.33, P<0.001)”.

Discussion section:

“Our study found higher odds for swine to be seropositive to IDV when compared with cattle”.

3)  Line 121 and in the table 2, the name Ivoire for the country is missing a capital letter.
Answer: This typo has been corrected.

4) The authors mentioned they collected the age of the tested animals but it was not taken into account as a factor. Would it be interesting to study despite the fact that the population tested was adult?

Answer: We thank the Reviewer for this suggestion. It would have been interesting to explore the relationship between the age of animals and the seropositivity rate of IDV. However, in our study, this would not be possible because we only differentiated animals in adults (cattle older than 2 years and small ruminants and swine older than 1 year) and young animals (cattle younger than 2 years and small ruminants and swine younger than 1 year) as it was difficult to get the accurate age of each animal. To make the sentence clearer in the materials and Methods section, as we have specified that all the animals were adults, we have removed the word “age” and the new sentence reads as follows (line 74):

……….”Animal health status and sex as well as the date and place of sampling were recorded”.

5) In the Discussion section, line 136, the authors may add "previously" to better understand the comparison of the studies from the same period: ...in Togo (6.3%) compared to data previously reported between 2017-2019 (4.5%)...
Answer: "previously" has been added.